# Histopathology based AI model predicts anti-angiogenic therapy response in renal cancer clinical trial

Jay Jasti[1], Hua Zhong[1,2], Vandana Panwar[2], Vipul Jarmale [1], Jeffrey Miyata[3], Deyssy Carrillo[3], Alana Christie[3,4], Dinesh Rakheja [2], Zora Modrusan[5], Edward Ernest Kadel III [6,7], Niha Beig[8], Mahrukh Huseni[6], James Brugarolas [3,9], Payal Kapur [2,3] ✉ & Satwik Rajaram [1,2,3] ✉

Anti-angiogenic (AA) therapy is a cornerstone of metastatic clear cell renal cell carcinoma (ccRCC) treatment, but not everyone responds, and predictive biomarkers are lacking. CD31, a marker of vasculature, is insufficient, and the Angioscore, an RNA-based angiogenesis quantification method, is costly, associated with delays, difficult to standardize, and does not account for tumor heterogeneity. Here, we developed an interpretable deep learning (DL) model that predicts the Angioscore directly from ubiquitous histopathology slides yielding a visual vascular network (H&E DL Angio). H&E DL Angio achieves a strong correlation with the Angioscore across multiple cohorts (spearman correlations of 0.77 and 0.73). Using this approach, we found that angiogenesis inversely correlates with grade and stage and is associated with driver mutation status. Importantly, DL Angio expediently predicts AA response in both a real-world and IMmotion150 trial cohorts, out-performing CD31, and closely approximating the Angioscore (c-index 0.66 vs 0.67) at a fraction of the cost.

The treatment options for patients with metastatic clear cell renal cell carcinoma (ccRCC) include anti-angiogenic (AA) therapies (e.g., vascular endothelial growth factor receptor-tyrosine kinase inhibitors VEGF-TKIs), immune checkpoint inhibitors (ICI), mammalian target of rapamycin (mTOR) inhibitors and, most recently, a hypoxia inducible factor (HIF)-2 inhibitor. These drugs are administered either alone or in combination[1]. However, none of these therapies uniformly benefit all patients. In addition, whether combinations, such as of ICI and AA, are synergistic is unclear[2]. In fact, recent biomarker analyses suggest that some ccRCCs are exclusively responsive to one or the other[3–5]. As a

result, drugs are being administered to patients that offer limited benefit, but have potential toxicity, and an associated financial burden. Thus, there is a critical need for predictive biomarkers of treatment response.

Multiple strategies have been deployed to identify biomarkers for ccRCC, but none have advanced to the clinic. Arguably, to date, the most promising approaches have used RNA sequencing (RNAseq). McDermott et al.[5] showed that patients with higher expression of angiogenesis related genes (Angioscore) exhibited better response to AA therapy in the phase 2 IMmotion150 trial. Similar analyses were

[1]Lyda Hill Department of Bioinformatics, University of Texas Southwestern Medical Center, Dallas, TX, USA. [2]Department of Pathology, University of Texas Southwestern Medical Center, Dallas, TX, USA. [3]Kidney Cancer Program, Simmons Comprehensive Cancer Center, University of Texas Southwestern Medical Center, Dallas, TX, USA. [4]O'Donnell School of Public Health, The University of Texas Southwestern Medical Center, Dallas, TX, USA. [5]Department of Proteomic and Genomic Technologies, Genentech, South San Francisco, CA, USA. [6]Translational Medicine Oncology, Genentech, South San Francisco, CA, USA. [7]US Medical Affairs, Genentech, South San Francisco, CA, USA. [8]gRED Computational Sciences, Genentech, South San Francisco, CA, USA. [9]Department of Internal Medicine (Hematology-Oncology), University of Texas Southwestern Medical Center, Dallas, TX, USA. ✉e-mail: payal.kapur@utsouthwestern.edu; satwik.rajaram@utsouthwestern.edu

performed in the phase 3 IMmotion151 and Javelin trials[4,6]. Notably, an RNA-based biomarker is now being explored prospectively in the OPTIC RCC trial (NCT05361720)[7].

However, transcriptomic-based biomarkers are challenging for everyday clinical use. Not only are they time-consuming but also suffer from experimental variability as readouts are susceptible to sample quality issues and batch effects[8–10]. Moreover, the assays are costly and typically only a small portion of the tumor is profiled, which may be problematic in notoriously heterogenous ccRCC[11–13] where multi-region sequencing shows that there is extensive variability[14].

Importantly, the Angioscore is largely based on genes expressed by endothelial cells. This is unsurprising since endothelial cells are responsible for the vasculature in tumors and are the targets of AA drugs, which act on VEGF receptors expressed on their surface. Endothelial cells are visually distinct in H&E slides and can be segmented using computational models in both ccRCC[15] and other cancers[16]. For reasons that are not completely understood, however, while immunohistochemical (IHC) staining of endothelial cells using CD31 correlates with the Angioscore, the relationship was weaker (spearman correlation = 0.62 in IMmotion150) than one might expect, possibly due to technical challenges associated with IHC quantification[5].

We hypothesized that Hematoxylin and Eosin (H&E) stained histopathologic slides may be used to directly quantitate angiogenesis. This would offer a more direct measurement of tumor vasculature and would address some of the limitations of the Angioscore, including cost and delays. Furthermore, inasmuch as H&E slides are used to render a diagnosis, they are typically obtained from multiple areas and capture tumor heterogeneity. Thus far, the approaches that have shown most promise in predicting gene expression signatures from H&E slides have been weakly supervised deep learning (DL) models that determine slide level expression by collating local patch level predictions, without reference to underlying cell types[17–24]. While it is possible to gain some understanding of the slide level predictions based on which patches have low or high gene expression, the lack of direct interpretability of the patch level predictions themselves poses a challenge.

Here, we present a visually interpretable DL biomarker for histopathological slide image analysis that correlates with the RNA-based Angioscore to infer response to AA therapy in ccRCC. A key feature is the use of both the RNA Angioscore and physical endothelial cells as training ground truth, which increases robustness and furthers interpretability. Specifically, the model generates a visual representation of the vascular network at the pixel level setting the foundation for predictions. We show that this predicted Angioscore from H&E images alone (H&E DL Angioscore) correlates strongly with the RNA based Angioscore in multiple independent cohorts including from the IMmotion150 clinical trial. Further, we explore the relationship of the H&E based Angioscore with various clinical and prognostic variables including grade, stage, and gene status. Finally, we validate the performance of the model as a predictor of response to AA therapy on a real-world clinical dataset and on data from the IMmotion150 clinical trial.

## Results

### Building an H&E based deep learning model to predict the RNA-based Angioscore

To predict the Angioscore, we developed a workflow that applies a DL model to tumor regions identified in whole slide images (WSI) of H&E-stained ccRCC slides (Fig. 1A, Methods). A key design principle was for the output of the model to be visually interpretable. To achieve this, we trained the model to predict a vascular network ("vascular mask" based on CD31 IHC). The final output of the model – termed as the H&E DL Angioscore – was intended to match the Angioscore from

RNA while being a simple summary of a visually interpretable prediction, the vascular mask (Fig. 1B). This approach makes it possible to visually interpret the basis of model predictions, allowing us to build confidence in its performance and diagnose deviations from expected trends. For example, on slides with intra-tumor heterogeneity, not only can we identify the areas with differing Angioscores (Fig. 1C shows a comparison to multi-region sequencing in select areas) but being able to visualize the vasculature underpinning these prediction provides a much greater degree of interpretability than simply knowing if a local patch had low or high expression (a synthetic averaging of our prediction at a patch level is provided in Fig. 1C for comparison).

To overcome the limitation that we did not simultaneously have RNA and CD31 IHC on the same samples, we built a mixed DL model that separately predicts the vascular mask and the RNA Angioscore and enforces consistency between these two predictions (Fig. 1B, C, Supplementary Fig. 1A, Methods). To build the CD31 model (i.e. vascular mask prediction), H&E and IHC images were computationally aligned, and pixels were given ground truth assignments as CD31 positive or negative based on an IHC model (Supplementary Fig. 2, Methods: UTSW CD31 Re-stain dataset, Supplementary Table 1). A U-Net[25] model with a Resnet[26] backbone was then trained to recover these ground truth positive/negative pixel assignments based purely on the H&E input. The RNA Angioscore prediction model shares the encoder portion with the vascular mask model but has its own subnetwork that predicts a single Angioscore value for each patch. This model was trained by using public data from the TCGA KIRC (Supplementary Table 1); for each slide image patches from the tumor regions served as input, and the matching RNA Angioscore[27] served as target ground truth. Finally, to establish consistency between the two arms, we required concordance between the predicted RNA Angioscore and the percentage of positive pixels from the CD31 prediction. To train our models we alternate between patches of data with CD31 and RNA ground truth, updating the network weights to maximize agreement with the appropriate ground truth as well as consistency between the Angioscore and CD31 predictions (Supplementary Fig. 1). Models were trained using 3-fold cross validation of the TCGA data (Supplementary Table 2), and we selected the best performing model (Fold 1) for all downstream analysis while the remaining two folds were used to assess result stability.

### Validation of H&E DL Angioscore model

We first tested the performance of our DL model on held-out portions of our training sets. We compared the endothelial cell outputs of the CD31 arm to the CD31 IHC (Supplementary Fig. 2) and found good segmentation performance with a tendency to overpredict the boundaries of the CD31 mask (Methods, precision = 0.53, recall = 0.66, F1 = 0.58, and Supplementary Fig. 3). In addition, an expert genitourinary pathologist (PK) reviewed the output masks in conjunction with the H&E images in the TCGA cohort to ensure that there were no systematic over- or under-predictions leading to deviations with the ground truth RNA Angioscore. Next, we tested the performance of our RNA Angioscore predictions. In principle, our model has two readouts that should correlate with the RNA Angioscore: the output from the RNA score arm and the percentage of positive pixels from the CD31 mask arm. We first confirmed that, although connected by a non-linear transform, these two readouts are strongly correlated (Supplementary Fig. 4 right column, Spearman correlation = 0.95). Since results would be unchanged by choice of readout, to maximize interpretability we use the CD31 mask arm for all further analyses. We henceforth refer to the percentage of positive pixels from this arm as the H&E DL Angioscore. Next, we compared the H&E DL Angioscore to the RNA Angioscore on the held-out portion of the training TCGA cohort (consisting of ~33% of the slides) and found a correlation of 0.68 (Fig. 2A).

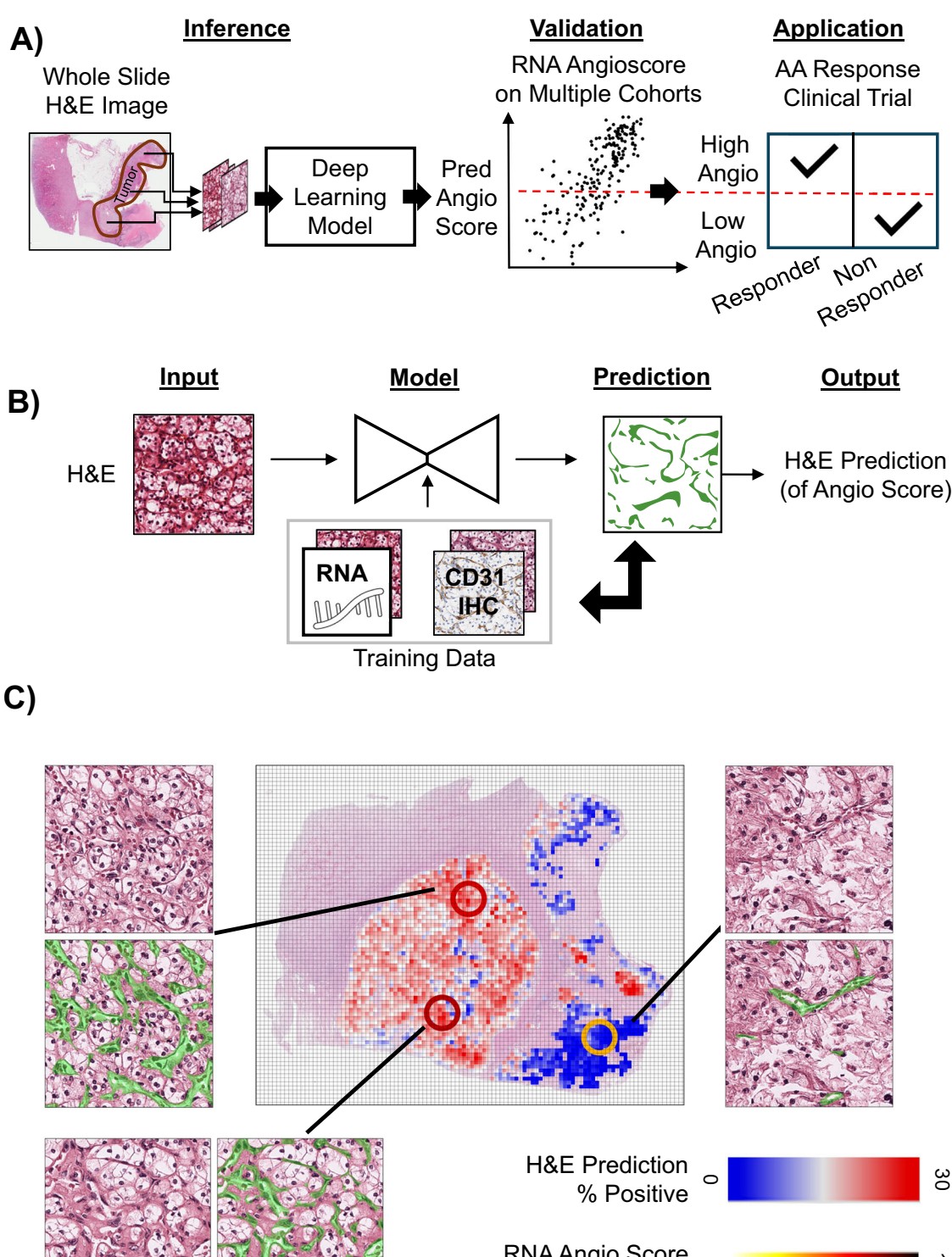

## Performance of H&E DL Angioscore model in predicting RNA-based Angioscore on real-world and clinical trial datasets

Next, we tested the ability of the H&E DL Angioscore to predict the true RNA Angioscore on two previously unseen independent cohorts (UTSeq and IMmotion150, Supplementary Table 1). Unlike the TCGA, where the RNA and H&E tissue are from different samples, and hence potentially impacted by intra-tumor heterogeneity, the tissue for H&E and RNA seq analysis are spatially matched in these two cohorts. We first tested the performance on the UT Sequencing cohort (UTSeq, Methods), which is a custom morphology guided sequencing dataset

with 196 samples. This UTSeq cohort has a tight alignment between morphology and sequencing as punched areas for sequencing analyses were flanked by matching H&E images on top and bottom. In this cohort, we observed a correlation of 0.77 between the RNA Angioscore and our predicted H&E DL Angioscore (Fig. 2B, Supplementary Fig. 5). Next, we tested the performance of our model on the IMmotion150 clinical trial slides (Fig. 2C). This cohort contains 226 available H&E slides each representing a single patient, and the RNA was extracted by macro-dissection of serial sections[5]. We obtained similar strong agreement (correlation of 0.73) between the H&E DL Angioscore and

**Fig. 1 | Project and approach overview. A** Schematic outlining the H&E DL Model development, validation and practical application of the model. The model predicts Angioscores directly from an H&E-stained slide and is validated against the RNA-based Angioscore using multiple independent datasets. The validated model is applied to independent, previously unseen, clinical datasets where its predicted Angioscore is correlated with response to antiangiogenic (AA) therapy. **B** H&E DL Model is an interpretable machine learning model to predict Angioscore from H&E images. Given an input H&E image, the model predicts a vascular mask (green), and the proportion of positive pixels is the output H&E-based Angioscore. Training data consists of two datasets with H&E images matched with RNA-based Angioscores and CD31 IHC (basis of the vascular mask), providing the target ground truth. The model is trained to predict the vascular mask matching the CD31 and the RNA-

based Angioscore (see Supplementary Fig 1 and methods for details). **C** Illustration of the model output on a ccRCC case with intra-slide heterogeneity and available multiregional RNA sequencing data. The central plot shows the model H&E Angioscore (blue-white-red colormap) applied to all tumor areas on the slide. To contrast against the output from patch level models (e.g., based on MIL), we calculated a local average of the percentage positive vascular mask prediction within a 416 x 416px grid, which show a far lower level of explainability than the pixel level vascular masks shown alongside. Three areas (circles) are marked where we had the ground truth RNA-based Angioscores (circle colors in yellow, red colormap). There is a broad qualitative agreement in the amount of vasculature and the H&E- based and RNA-based Angioscores, and both capture ITH which would have been missed by the standard slide-level bulk sequencing approaches.

## A) TCGA Holdout
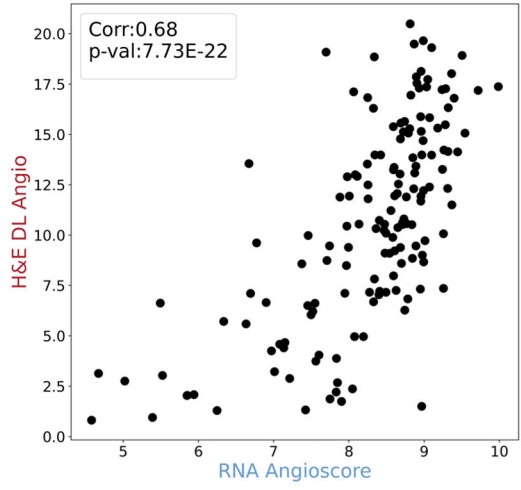

## B) UTSEQ
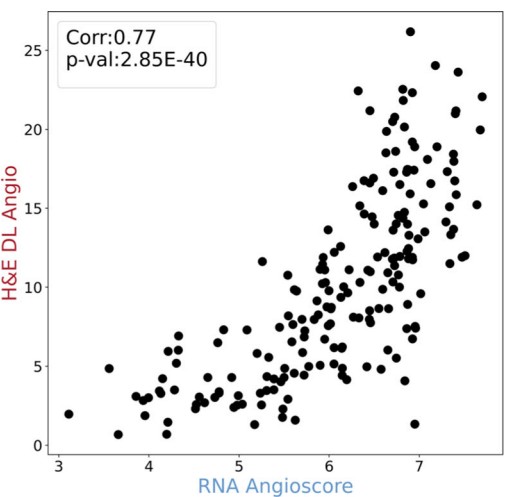

## C) IMmotion150
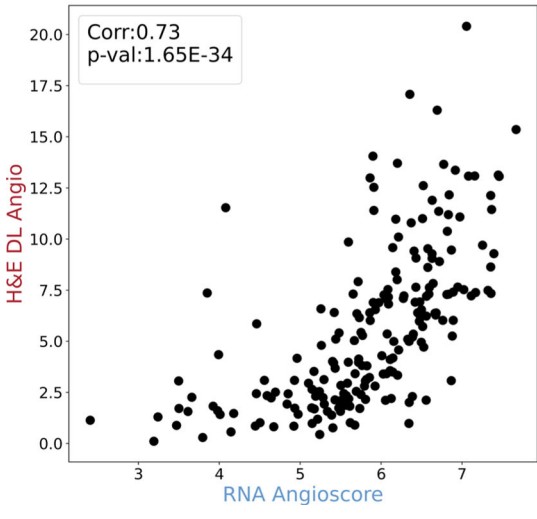

## D) IMmotion150: CD31
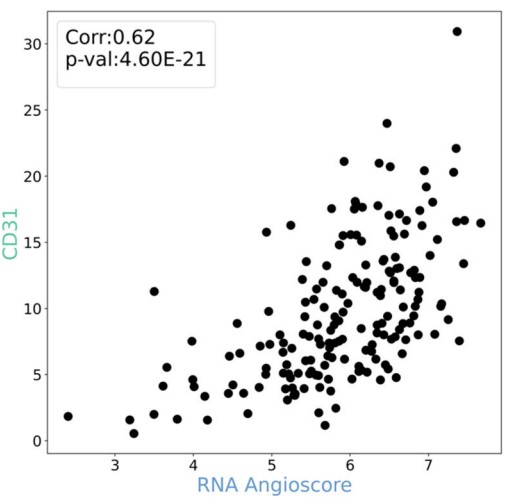

**Fig. 2 | H&E DL Model reliably predicts RNA-based Angioscore across multiple cohorts.** Each panel shows a scatter plot (each point represents a sample) comparing RNA-based Angioscore (x-axis) and predicted scores (y-axis). Spearman correlation coefficient along with the *p*-values are displayed in the legend. All *p*-values are calculated using two-sided Spearman rank correlation test. **A** Model Performance on TCGA held-out dataset (*N* = 154). **B** Model performance on independent UTSeq dataset. Predictions were averaged in cases where 2 slides

representing the top and flip sides of the block were available (*N* = 196). **C** Model performance on IMmotion150 dataset (*N* = 200). **D** Correlation between CD31 measurements and Angioscore measurements for the IMmotion150 dataset shows that model predictions (i.e., panel **C**) correlate better with RNA-based Angioscore and outperform CD31 measurements (*N* = 184). Source data are provided as a Source Data file.

the RNA-based score for the IMmotion150 cohort. Importantly, our model performs well (Supplementary Fig. 6) regardless of tissue extraction site (primary vs metastatic) as well as procedure (resection vs biopsy). Moreover, our overall performance remained stable across different fold models (UT Seq correlation = 0.77/0.76/0.74, IMM150 correlation = 0.73/0.73/0.72) as shown in Supplementary Table 2. Additionally, the mean and standard deviation values confirm the model's stability across the folds (Supplementary Table 2). Notably, in both the IMmotion150 (Fig. 2D, correlation of 0.73 vs 0.65) and the subset of UTSeq samples with both RNAseq and CD31 IHC (Supplementary Fig. 7, correlation of 0.61 vs 0.45), the H&E DL Angio correlates much better with RNA Angio than the CD31 IHC does.

To better understand the strong performance of our model we tested alternate architectures trained on our datasets. The traditional approach[19,23,24] uses a pretrained network to extract features from a large number of patches in a slide, and multiple instance learning (MIL) to train a simple network to make a slide level prediction from these features. Our requirement of pixel level predictions for interpretability led to three major differences: a) the addition of a CD31 segmentation arm, b) end-to-end training of the full network, and c) (because the entire slide cannot be held in memory) predicting the slide level Angioscore based on a small number of patches per training batch (BatchMSE; Supplementary Fig 1B). To interpret these factors, we trained models using: a) MIL-Regression (based on CAMIL[23]) trained using RNA as ground truth, b) models with our BatchMSE approach with RNA only as ground truth but with different parts of the network held frozen, and c) a segmentation model trained using CD31 only as ground truth. Across the three datasets, for the RNA trained models, there was no clear-cut advantage to either a large set of training set of patches in a bag or end-to-end training, but our mixed model consistently outperformed the models trained on a single modality (Supplementary Table 3).

## Exploring H&E DL model Angiogenesis prediction for biomarker discovery

Having validated our H&E based model's quantification of angiogenesis, we asked how its predictions correlated with well-established prognostic variables. We leveraged our previously published Tissue Microarray (TMA) cohort with over 800 punches that otherwise lack gene expression data[28] (Methods). We found an inverse correlation between the World Health Organization/ International Society of Urological Pathology (WHO/ISUP) nucleolar grade and the H&E DL Angioscore (Fig. 3A). We extended these analyses to the cohorts where we have RNA and observed a similar inverse relationship between grade and Angioscores based on RNA or H&E (Supplementary Fig. 8). Interestingly, the H&E DL Angioscore for a given grade is more consistent across cohorts than when derived from RNA, suggesting our model could provide a means to overcome batch effects that impact transcriptomic signatures and pose a significant challenge for clinical adoption.

Extending our analysis to TNM stage (Fig. 3B), we found that tumors with high stage (stage 3 and 4) are associated with lower angiogenesis than low stage ccRCCs (stage I and II). Similarly, other prognostic factors like tumor size (Supplementary Fig. 9A) and presence of sarcomatoid features (Supplementary Fig. 9B) negatively correlated with the H&E DL Angioscore. Taken together these data suggest that angiogenesis is progressively reduced with tumor progression. Next, we evaluated the H&E DL Angioscore across ccRCC architectural subtypes we previously reported that correlate with patient outcome[11]. In the UTSeq cohort, we found the expected[11,29,30] reduction of angiogenesis (both on RNA and H&E DL Angioscore) with more aggressive architectural patterns (Supplementary Fig. 10). Overall, our results show that our DL model accurately captures vascular network, and that vascular network is strongly associated with tumor architecture, as well as tumor grade and stage.

We have previously shown that two genes that are frequently mutated in ccRCC, *PBRM1* (50%) and *BAP1* (15%), are associated with tumor grade and aggressiveness (44). Furthermore, by engineering mice with mutations in these genes, we showed that mutations in these genes drive tumor grade (39). Specifically, while Pbrm1-deficient ccRCC were of low grade, Bap1-deficient ccRCC had high grade. Given the correlation of grade and stage with frequently mutated driver genes in ccRCC[28], we sought to capture the effect of BAP1 and PBRM1 loss on angiogenesis. BAP1 loss correlated with lower H&E DL Angioscores relative to wild-type (4.9 vs 9.4, $p = 3.67 \times 10^{-7}$), whereas PBRM1 loss led to a slight (9.9 vs 9.4) not-statistically significant increase (Fig. 3C).

Next, we tested the extent to which the H&E DL Angioscore correlated with survival. As might be expected based on our previous results, there is a strong relationship between overall survival and the H&E DL Angioscore with a c-index of 0.75. To visualize this relationship using the Kaplan–Meier method, we stratified patients based on their H&E DL Angioscore. To avoid overfitting on this dataset and assess robustness, we used an independent dataset (from TCGA), to establish cutoffs (Supplementary Fig. 11). We observed two peaks in $p$-value, one that separated out all the highly angiogenic samples with good prognosis and another that sequestered the low Angioscore samples with unfavorable prognosis (Supplementary Table 4). These cutoffs also translate to other cohorts separating out our previously described indolent, intermediate, and aggressive architectural patterns[11] in the UTSeq cohort (Supplementary Fig. 10A). Based on these observations we applied the low and high peak thresholds from the TCGA to the TMA cohort and performed a three-class stratification (Fig. 3D) which provides HR values of 6.9 (3.7–12.8) and 2.9 (1.7–4.9) for the high and medium DL Angio groups against the low, suggesting that the H&E DL Angioscore can effectively stratify patients with different outcomes.

## H&E DL Angioscore predicts AA therapy response

Given that the RNA Angioscore is associated with response to AA therapy, we next sought to test the predictive performance of our H&E DL Angioscore. First, we applied our model to a real-world cohort consisting of 145 patients treated at UTSW who received single agent first line AA for metastatic ccRCC between 2006 to 2020. Using Time-to-next treatment (TNT) as a proxy for treatment efficacy we found a c-index of 0.6 with the H&E DL Angioscore. Next, we performed Kaplan–Meier and Cox-proportional hazards analyses by stratifying the patients based on their H&E DL Angioscore. As this cohort was restricted to metastatic cases (which as expected had few cases with higher Angioscore), we performed a two-class stratification to identify low angiogenic tumors using the lower of the two thresholds described above (Fig. 4A; threshold H&E DL Angioscore of 5.66) and obtained a HR of 0.64 (95% CI: 0.45–0.91) with a $p$ value of 0.012. Interestingly, the best stratification (with hazard ratio of 0.41; 95% CI: 0.26–0.64; $p$ value of $8.7 \times 10^{-5}$) was obtained using an even lower Angioscore threshold of 2.34, likely indicating that this cohort has a greater proportion of aggressive tumors with low angiogenesis that may not benefit from AA therapy.

Finally, we tested the potential of our H&E model to predict AA therapy response, using a carefully curated clinical trial dataset (IMmotion150). This trial contains three treatment arms, a TKI (sunitinib), an ICI (atezolizumab), and the combination of atezolizumab with an anti-VEGF antibody, bevacizumab. We compared the predictive value of the H&E DL Angioscore to that of the previously reported RNA Angioscore as well as CD31 IHC for response to sunitinib. First, we examined the relationship to progression free survival (PFS) and found c-indices of 0.66 (H&E DL Angioscore), 0.67 (RNA Angioscore) and 0.55 (CD31 IHC). Moreover, the performance of the H&E based model was stable across folds with c-indices of 0.66, 0.63 and 0.64. Notably, our model (and the RNA Angioscore) greatly outperformed standard clinical measures like grade and factors such as

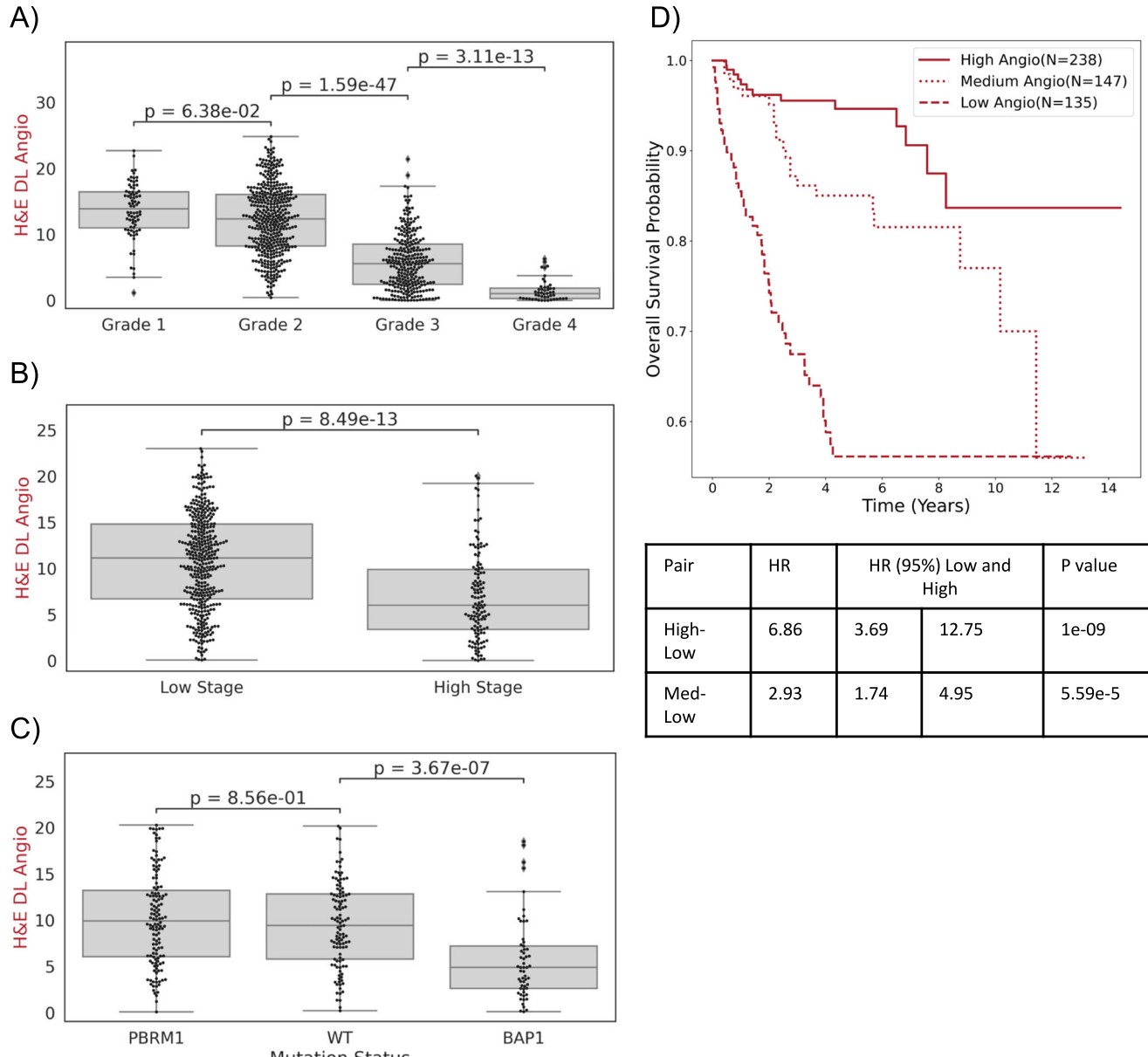

**Fig. 3 | Model predictions correlate with known prognostic variables in independent datasets.** The H&E DL Angioscore model was applied to independent tissue microarrays and its output was compared to various prognostic variables including **A** Nuclear Grade ($N = 76$ Grade 1, $N = 409$ Grade 2, $N = 278$ Grade 3, $N = 48$ Grade 4), **B** Patient Stage ($N = 390$ Low Stage, $N = 130$ High Stage) and **C** Functional status of driver genes *BAP1* and *PBRM1* (the few cases with loss of both *BAP1* and *PBRM1* are considered as exhibiting BAP1 loss)($N = 134$ *PBRM1*, $N = 113$ *WT*, $N = 57$ *BAP1*) *P*-values for Figs. **A**–**C** are calculated using Mann–Whitney U test, a non-parametric two-sided test. Box plots in (**A**–**C**) show median values with inter quartile ranges (IQR) and the whiskers show 1.5 times IQR values. **D** Kaplan–Meier curves showing overall survival of 520 patients stratified by H&E DL Angioscore (c-index = 0.75). Threshold scores for stratification were independently determined from TCGA overall survival dataset as 10.3 and 5.6. Note: *p*-values are calculated using two-sided non-parametric log rank test. Source data are provided as a Source Data file.

the MSKCC[31] that are typically considered in making treatment decisions (Supplementary Table 5). Next, we stratified patients into low/high angiogenesis groups based on the median score for each assay following the original IMmotion150 publication (results are even stronger for H&E DL Angioscore with the TCGA based cutoff above; Methods, Supplementary Table 4). We generated Kaplan–Meier curves and performed Cox-proportional hazards calculations (Fig. 4B, Supplementary Fig. 12), which showed that the RNA and H&E based predictions of sunitinib response are comparable and far superior to the CD31 IHC. This point was further reinforced by separate analyses comparing our three assays in terms of: a) the fraction of patients who responded to sunitinib among high/low Angiogenesis groups

(Fig. 4C), and b) the AUC in predicting the objective response (responder or not) based on Angiogenesis (Fig. 4D). Additional information on sunitinib treatment response and treatment response to other drugs used in IMmotion150 trial are shown in Supplementary Figs. 13, 14 respectively. Interestingly, our score (like the RNA based score) captures the previously reported[32,33] inverse relationship between angiogenesis and response to ICI (Supplementary Fig. 14, relative heights of bars across arms). Taken together, our results show that the DL based model predicting the Angioscore solely from H&E images rivals the gold-standard RNA-based Angioscore and greatly outperforms the CD31 IHC in both real-world and clinical trial datasets (IMmotion150).

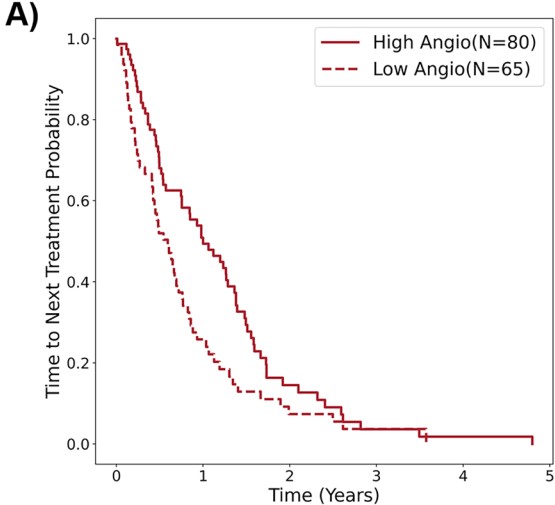

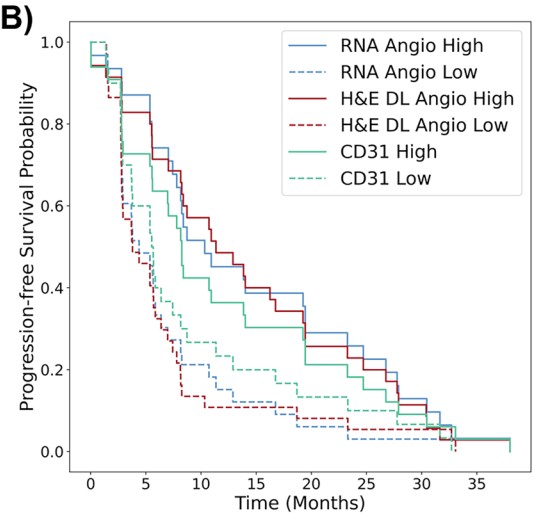

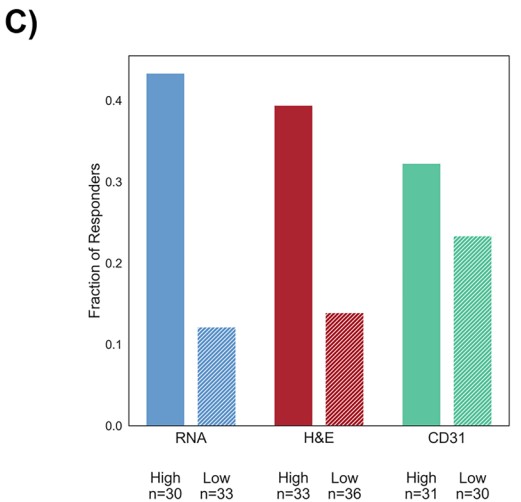

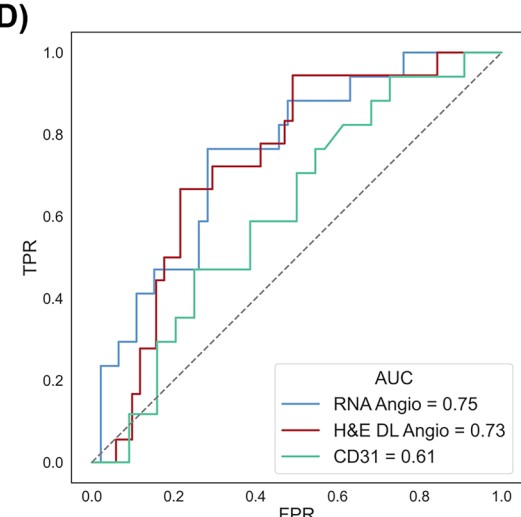

**Fig. 4 | H&E DL Angioscore predicts response to Anti-Angiogenic therapy.**
**A** Kaplan–Meier analysis of H&E DL Angioscore-based stratification of the AA response of patients from the UTSW-Clinical cohort. Time to next treatment (TNT, x-axis) was used as a measure of treatment response, and patients were stratified into Low and High DL Angioscore based on the threshold (5.6) determined in TCGA prognosis to perform a Cox-Proportional hazards analysis (c-index = 0.6, Hazard ratio (Hr): 0.64 (95% confidence interval (CI): 0.45–0.91), *p* value: 1.23e−2) *p*-values are calculated using two-sided non-parametric log rank test. **B** Kaplan–Meier analysis comparing RNA Angioscore, H&E DL Angioscore and CD31 in their ability to stratify the Sunitinib response of patients from the IMmotion150 clinical trial. Progression free survival (x-axis) was used as a measure of treatment response.

Patients were stratified into Low and High groups based on the median levels of their corresponding H&E DL Angioscore and RNA/CD31 Angioscores as in the IMmotion150 trial. **C** Comparison of the fraction of patients who responded to Sunitinib treatment among the low and high angiogenesis groups as determined by RNA Angioscore, H&E DL Angioscore and CD31 (number of patients in each group specified in graph). **D** AUC curves comparing how the RNA Angioscore (*N* = 56), H&E DL Angioscore (*N* = 65) and CD31 (*N* = 57) can distinguish Sunitinib responders and non-responders in the IMmotion150 clinical trial. Responders were patients having complete or partial response, and non-responders are those with stable or progressive disease. Source for (**A**, **C**, **D**) are provided. PFS data in 4B cannot be shared in order to ensure the highest patient-level privacy.

## Discussion

Anti-angiogenic (AA) drugs are used to treat metastatic ccRCC, either as monotherapy or in combination with ICI. Recent data suggest that ccRCC with high levels of vascularity respond better to AA therapy. Indeed, in both IMmotion150 and IMmotion151 trials, high expression of a 6-gene Angioscore signature that included *PECAM1* (gene coding CD31) was associated with improved progression free survival (PFS) in patients treated with the AA agent sunitinib[5,6]. These results from IMmotion150 and IMmotion151 were confirmed by analysis from the JAVELIN renal 101 trial[4] and have led to a prospective clinical trial where RNA expression data is used to determine therapy.

However, clinical adoption of transcriptomic gene signatures is challenging due to difficulties in standardization, delays, and high cost. Furthermore, the results are confounded by the notorious heterogeneity of ccRCC tumors[32,33]. Indeed, many ccRCCs harbor areas of both low and high angiogenesis[14]. To address this, we present a robust deep learning (DL) model that predicts the Angioscore from H&E-stained slides, offering a scalable and cost-effective alternative to RNA-based assays. Notably, the output of our model was strongly predictive of the RNA Angioscore on both unseen real-world (Spearman correlation, 0.77 for UTSeq) and clinical trial datasets (Spearman correlation, 0.73 for IMmotion150).

To maximize interpretability and robustness, our H&E based DL model was trained using both RNA and CD31 IHC, which offers advantages over each assay. Indeed, H&E DL Angioscore surpasses the predictive capabilities of CD31 IHC in predicting both the gold standard RNA Angioscore and treatment response. Future studies are needed to determine if this is purely due to technical challenges in performing and quantifying CD31 IHC, or whether the combined training with RNA leads to biologically meaningful differences in the vascular masks. However, the inclusion of CD31 IHC did not increase the performance of H&E DL Angioscore compared to the gold standard, the RNA Angioscore, in prediction of response. Importantly, our model offers a more standardizable measurement across diverse samples mitigating the impact of RNA batch effects. As H&E slides are ubiquitous, it is feasible to profile multiple areas within heterogeneous ccRCC tumors, and our results make a compelling case for pursuing clinical biomarkers based on the application of DL to histopathology slides. In addition, the H&E DL Angioscore can be applied in situations where tumor tissue is limiting (biopsies), where RNA extraction may not be possible.

An important distinguishing aspect of our DL model is that by training on both RNA and CD31 we can provide visually interpretable predictions and thereby overcome the "black box" limitation. This is critical for quality evaluation and clinical adoption. Alsaffin et al.[19] use a transformer-based DL model to predict bulk RNA sequence scores for about 30,000 genes from kidney cancer. In multiple cancers, previous efforts have demonstrated prediction of specific genes and pathways from H&E images, typically using MIL-Regression type approaches[19,23,24]. While patch level predictions can provide some insights as to why a slide was deemed to have high expression, the basis of the patch level predictions themselves are challenging to decipher. Beyond interpretability, our results also suggest that our multimodality approach improved the overall predictive power compared to single modality, and particularly an MIL approach based on RNA only (although it is conceivable that other pretrained encoders could perform better). One advantage of MIL is that, since it uses pre-extracted features, it has access to the full slide during individual batch updates. In contrast, because our end-to-end trained model needs more memory, we only sample 8 patches per batch to represent the slide. On the other hand, the distribution of endothelial cells may be uniform enough to be captured by a small random sampling of patches, so that the end-to-end training is able to extract more relevant image features. In terms of identifying cell types in H&E images, this has been demonstrated by multiple studies for immune cells, but there are few that connect this quantification directly to response prediction without inclusion of additional assays[34–36]. In ccRCC, classic image analyses (as opposed to deep learning) can identify endothelial cells, but with lower fidelity[15]. The ability of such models to predict AA response has not been tested. However, endothelial cells may not be enough, as our model outperformed the CD31 IHC in predicting both the RNA Angioscore and ultimately response.

In addition to AA therapy, patients with metastatic clear cell renal cell carcinoma (ccRCC) are often treated with ICIs, which may be given in combination with AA. However, it is unclear whether ICI and AA are truly synergistic[37] and there is evidence suggesting that tumors respond predominantly to one or the other[3–5]. Indeed, our data on the IMmotion150 trial suggests that although high-Angioscore patients are more likely than low-Angioscore patients to respond to sunitinib, the opposite is true for the arms including ICI.

Our H&E DL Angioscore has applications beyond predicting response. Angiogenesis is considered a hallmark of cancer[38], and is central to ccRCC biology, playing an important role in the definition of molecular subtypes[6]. H&E DL Angioscore enabled an in-depth exploration of biological phenomena, leveraging the vast amounts of archival tissue where we lacked transcriptomic profiling. The H&E DL Angioscore allowed us to explore the relationship of angiogenesis with various prognostic variables on a large cohort without the need for RNA-seq analysis. Our analysis shows that angiogenesis is inversely correlated with indicators of aggressiveness and disease progression including grade, presence of sarcomatoid elements, and tumor stage. In addition, it extended our previous findings on architectural patterns by showing that aggressiveness and vascularity inversely correlate[11]. It also supports findings from our human and genetically engineered mouse models showing that BAP1-deficient ccRCC are less vascular than PBRM1-deficient tumors[39]. Importantly, the H&E DL Angioscore provides a platform to operationalize and objectivize these analyses, which could be quite helpful to integrate multiple datasets.

Our study showcases the potential of H&E-based biomarkers for clinical application. A commonly held notion is that a c-index of 0.7 or greater is required for clinical applicability[40], and both our model and the RNA Angioscore fall marginally short (c-index = 0.66/0.67). However, RCC recurrence models commonly used in clinical practice such as TNM staging and Memorial Sloan Kettering Cancer Center (MSKCC) criteria have failed to meet this threshold in the past[31]. Indeed, at least in the IMmotion150 cohort, our model's performance is better than MSKCC criteria[1] (c-index = 0.53, Supplementary Table 4). There are many ways to further improve performance within the current framework. Increasing training data across diverse cohorts will enhance reliability and robustness to variations in staining or microscopes. The current model is only applied to tissue from the tumor regions in a slide and thus improving region identification models could improve the model's performance. Additionally, although our models perform well on biopsies, it is conceivable that small tissue samples may benefit from serial sections. Further, clinical biomarkers are typically composites of multiple features and based on results showing synergy between RNA biomarkers and other readouts in the IMmotion151 clinical trial[6], we expect that including our score in a nomogram will lead to significant improvements. Nevertheless, a first step towards any clinical adoption would require H&E DL Angioscore validation on data from additional clinical trials with AA therapy.

We anticipate future studies will examine the potential of H&E images to improve response prediction beyond the RNA Angioscore by a) considering the variation of Angioscores (rather than a single average level) across multiple H&E slides from a tumor to account for intra-tumor heterogeneity, b) directly predicting response from images without using the RNA Angioscore as an intermediate target, c) building composite biomarkers that include not only vascularity, but other relevant features of the tumor microenvironment (e.g. immune cells) and of tumor cells (e.g. driver mutation status) that can be predicted from H&E images. Integrative analyses may also include non-H&E biomarkers. Additionally, multi-modal approaches that combine different modalities such as pathology, radiology and genomics will likely result in a more comprehensive biomarker. We also expect this approach can be refined for predicting responses to AA specific agents or adapted to predict immunotherapy responses.

In summary, the H&E DL Angioscore enables prediction of anti-angiogenic therapy response in ccRCC solely from H&E images, offering a cost-effective and interpretable alternative to RNA-based assays. By bridging the gap between molecular insights and clinical feasibility, our work sets the stage for transformative advancements in ccRCC therapeutics.

## Methods
### Ethics statement
We analyzed patient data from several institutions in this study. TCGA did not require formal ethical approval for the study of anonymized samples. Analysis of UTSW samples was conducted with approval by our institutional (UT Southwestern Medical Center [UTSW] and Parkland Hospital, Dallas) Review Board (IRB: STU 022015-015). WSI and associated data from the Immotion150 trial was shared by Genentech via MTA (MMTA202102-0219).

## Cohorts

We made use of several cohorts, each consisting of whole slide images of formalin fixed paraffin embedded (FFPE) H&E slides scanned at either 20X (-0.5 microns per pixel) or 40X (-0.25 microns per pixel). Our model was based on 20X images, so all 40X slides were down sampled by a factor of 2. A high-level summary of all the datasets used in this study is presented in Supplementary Table 1

1.  TCGA KIRC dataset: This is the primary training dataset for the RNA prediction. It was downloaded from NIH GDC Data Portal[27,41,42] and consists of 519 whole slide images and associated data (transcriptome, patient-related data such as overall survival). Several slides were excluded from our analysis: 14 exhibited frozen sample-like artifacts, 5 had non-ccRCC-like pathology, 2 had only benign renal parenchyma, 2 slides had imaging/staining artifacts, 6 slides were duplicates from the same patient, and 28 lacked RNA information in the pan-cancer dataset. The remaining set of 462 slides was split 2:1 for model training and validation purposes. Majority of TCGA images are available in 40X magnification, and a small fraction are available at 20X.
2.  UTSW CD31 Re-stain dataset: This is the primary training dataset for the vascular mask prediction. It consists of 17 slides that capture the spectrum of ccRCC morphologies, displaying a range of grades, tissue architectures and vascularity. These slides were first stained with H&E and imaged at 20X magnification using an Aperio scanner. Next, the slides were de-stained, re-stained with antibody for CD31 (clone JC70A; Agilent CA) and re-imaged. Of the 17 slides, 13 were used for training and 4 were held out for testing.
3.  UTSW Multiregional sequencing dataset (UTSeq Data): This dataset was used to validate the H&E DL Angioscore predictions as well as predicted vascular masks (on a small set of samples with CD31 staining). It consists of 161 H&E-stained slide pairs from 27 patients. Multiple samples (punches) were taken from the same patient and RNA measurements with transcriptomic profiles were obtained. Each FFPE punch had flanking top and the bottom H&E-stained images. Serial sections were taken from a smaller set of 36 samples, and they were stained with CD31 and imaged. Most of this dataset was imaged at 40X (H&E), with a smaller fraction was imaged at 20X (CD31 IHC).
4.  IMmotion150 Dataset: This dataset was used to validate the Angioscore and TKI response predictions as well as compare them to CD31. From the 305 patients enrolled in the IMMotion150[5], we had access to 239 whole slide (WS) digital images of H&E-stained slides out of which 13 were excluded due to duplicates ($n = 5$) or quality issues (significant artifacts resulting in extremely low evaluable tumor fraction). The remaining 226 WS images were used along with the associated information on drug response data, RNA and CD31 information as described previously[5]. Most of the study was restricted to patients treated with sunitinib, and we analyzed all patients with available data for each assay type (H&E:69, RNA: 63, CD31: 61).
5.  UTSW TKI Response Dataset: This dataset consists of 145 H&E-stained slides taken from patients undergoing Anti-VEGF treatment. The Kidney Cancer Explorer - an IRB-approved data portal at UTSW with clinical, pathological, and experimental genomic data – was queried for any patients who received first line VEGF-I for metastatic renal cell carcinoma between 2006 to 2020[43]. The medications included in the search were axitinib, bevacizumab, cabozantinib, cediranib, pazopanib, sorafenib, sunitinib, and tivozanib. 355 patients were found who met these criteria, out of whom 180 had H&E-stained images already available to us for analysis. Data for patients whose treatment was stopped due to toxicity was excluded and the final dataset consisted of 145 patients.
6.  Tissue Microarray (TMA): This dataset is a combination of TMA datasets that were described previously[28]. Most patients were represented by multiple TMA punches. Grade information is available at the punch level (811 punches) whereas other information (Tumor size, stage, overall survival, sarcomatoid status) was available at patient level (520 patients). BAP1 and PBRM1 protein status (as assessed by IHC[44]) was available for a subset of 304 patients.

## RNA Angioscore calculation

For the UTSEQ data, the raw transcriptome sequencing data was processed by the SCHOOL[45] with human reference genome version GRCh38.86 and Fragments Per Kilobase of transcript per Million mapped reads (FPKM) genes were generated. FPKM was normalized to Transcripts Per Kilobase Million (TPM), then log-transformed with 1 added to avoid taking log of zero. For other cohorts, namely TCGA and IMmotion150, TPM data was directly obtained. The signature genes for determining Angioscore are *VEGFA, KDR, ESM1, PECAM1, ANGPTL4* and *CD34* following the IMmotion150 and IMmotion151 studies[5,6]. The Angioscore for each tumor sample was computed by the mean log transformed TPM of the Angio signature genes.

## Training data generation

The H&E DL Angio model simultaneously predicts, from H&E images, a CD31 trained "vascular mask" and the RNA Angioscore. The training data for each of these predictions was generated as follows.

**CD31 training data.** Ground truth data from the UTSW CD31 Re-stain cohort consisted of 416x416px H&E patches with matching "vascular" masks of the same size, with each pixel assigned as CD31 positive or negative as follows:

1.  Generation of aligned H&E and CD31 IHC image patches: H&E and CD31 stained slides were registered using a two-step process: an initial manual rigid registration at the slide level performed in QuPath (version 0.4.3)[46] to align the slides, followed by an automatic non-rigid registration at the local image level to align the shared hematoxylin channels of the IHC and H&E images. Non-rigid registration was done using patches of 512 × 512 pixels with SimpleElastix's multi-resolution, pyramid registration framework (version 2.0.0rc2)[47]. Each image pair goes through affine registration first and then deformable registration using B-splines. A smaller 416 × 416 pair of patches were extracted from the center of the registered image pair to remove any edge effects.
2.  Binarization of CD31 IHC: We trained a classifier to identify the CD31 positive areas in IHC-stained images and distinguish them from CD31 negative and non-specific/artifactual DAB staining. Specifically, we generated IHC images with manually annotated ground truth assignments of CD31+/CD31−/artifact and trained a U-Net based model. We validated the performance on the classifier on 20 similarly constructed IHC image, ground truth mask pairs from 4 slides that were not used in the training of the model (Supplementary Fig. 2) and then used it to generate the ground truth vascular mask for our H&E classifier. This model was then applied to the IHC patches in our CD31 Training data to generate a binary mask with pixels classified as CD31 positive or negative (pixels classified as artifact were treated as exhibiting negative staining).
3.  Data Split: 17 slides (with matched H&E and CD31 IHC) were split into 14 for training and 3 for model evaluation. Patches were extracted from tumor regions and registered as described above. In all, 41,918 patches were used in training the CD31 Model, while 5223 patch pairs were used for evaluation.

**RNA training data.** Ground truth data was extracted from the TCGA KIRC dataset. It consists of 416x416px H&E image patches with associated RNA Angioscore (calculated as described above, for the patient from whose slide the patch was extracted).

1. Patch Generation: We first identified tumor regions of slides, with preliminary identification based on a CNN model as described previously[28], followed by manual refinement by an expert pathologist (PK) as needed. We then sought to extract 1500 randomly placed patches within the tumor region of each slide. Note: To prevent oversampling the same areas in slides with limited tumor content, we established a threshold sampling density and in 3% of cases we extracted fewer than 1500 patches to stay within this limit.

2. Data splits: The 462 H&E-stained slides from the TCGA cohort were split 2:1 for model training and validation. This split resulted in 360,999 patches for training and 183,894 patches for validation. Each patch was assigned an RNA Angioscore such that all patches from the same slide have the same Angioscore. We have trained models for each split and used the best performing model.

## Model

Given an input image the model produces a segmentation mask and a numerical value for the Angioscore and tries to ensure their consistency. This is achieved by using a shared Resnet18 encoder which splits into three sections (Supplementary Fig. 1A). First, the mask prediction arm which takes in an H&E image patch and outputs a predicted vascular mask with two classes (CD31+/−) has a U-Net architecture with an ImageNet pre-trained ResNet-18 backbone. Second, the Angio score prediction arm, which takes as input an H&E image patch and outputs a single number (the Angio score) shares the encoder arm of the U-Net, followed by several convolutional layers. Finally, the consistency arm, takes the output of the mask arm (namely a binary activation mask), calculates the fractional CD31 positive activation (global average pooling), and performs a learnt non-linear transformation to predict the RNA output from the mask output.

**Model training.** For any training patch, as only a single type of ground truth (CD31 Mask or RNA) is available, we developed custom loss functions and training strategies:

1. Batching: The model is trained in batches composed of patches with RNA (TCGA) or CD31 ground truth which have batch sizes of 32 and 4 patches respectively. ResNet encoder has batch norm layers and through experimentation we found that best performance can be achieved by forcing the batches to be sampled from different slides instead of randomly sampling patches combined from all slides. Each RNA batch of 32 patches is sampled from 4 different slides (thereby allowing us to average the slide level predictions across 8 patches, while also stabilizing the Batch Normalization calculations by sampling multiple slides per batch) while the 4 patches for CD31 are selected randomly. RNA batch sampling is further illustrated visually in Supplementary Fig. 1B.

2. Loss Functions: We use a different combination of loss functions for the two types of batches:

a. RNA Ground Truth: The predicted Angio scores $P^{Angio}$ and $P^{Cons}$ from the Angio and Consistency arms respectively are each averaged across patches from the same slide and compared to the true RNA Angioscore $T^{Angio}$ using a batch mean square error

$$L_{\text{RNA}} = \frac{1}{N}\sum_{S \in B}\left(\left\|T_S^{\text{Angio}} - \frac{1}{N_S}\sum_{p \in S}P_p^{\text{Angio}}\right\|^2 + \left\|T_S^{\text{Angio}} - \frac{1}{N_S}\sum_{p \in S}P_p^{\text{Cons}}\right\|^2\right) \quad (1)$$

Where $S \in B$ indicates the slides $S$ present in the batch, $N = 4$ the total number of slides in the batch, $p \in S$ denotes the patches $p$ in the batch belonging to slide $S$, and $N_S = 8$ denotes the number of patches from slide $S$ in the batch.

b. CD31 Mask Ground Truth: The loss is a sum of i) a segmentation loss $L_{Seg}$ between the true CD31 masks $T^{Mask}$ and the predicted mask $P^{Mask}$ and ii) a consistency loss $L_{Cons}$ comparing the

predicted Angio predictions from the RNA and Consistency arms as follows:

$$L_{Seg} = 0.9 \times \text{Dice}\left(T^{Mask}, P^{Mask}\right) + 0.1 \times \text{WCCE}\left(T^{Mask}, P^{Mask}\right) \quad (2)$$

$$L_{Cons} = \text{MSE}\left(P^{Angio}, P^{Cons}\right) \quad (3)$$

Where Dice denotes the Dice loss, WCCE is a class-weighted categorical-cross entropy and MSE is the mean square error.

3. Augmentation: During training we augmented images using mirroring and color augmentation HED adjust[48] with parameters (0.975,1.025). We also reduced saturation of the input images by a multiplier randomly chosen in the range of [0,0.5] since some of our cohorts have highly faded slides.

4. Training: We first pre-trained the mask prediction arm (i.e. the model without the RNA and Consistency arms) with the segmentation loss only, for 10 epochs with Adam optimizer with a learning rate of $10^{-4}$. The full mixed model was then trained for 10 epochs using stochastic gradient descent optimizer with initial learning rate of $10^{-4}$ and a momentum value of 0.9. Model was stored at every epoch and the best model was selected as the one having the highest Spearman correlation coefficient with the TCGA held out dataset (Supplementary Table 6).

## Model inference.

1. Stain Normalization: To reduce the impact of slide color variations, prior to model inference, we normalized slides from all cohorts (except the TCGA holdout) to match the color distribution of the training TCGA cohort as outlined previously[28].

2. Patch Generation: Patches were selected randomly from tumor areas as described for the training cohort. In the case of the cohorts where whole slides were profiled (TCGA Holdout, IMmotion150 and UTSW TKI) we targeted 1500 patches per slide as in the training cohort. However, for the UTSeq and TMA cohort where local regions were profiled, we targeted 250 and 1000 patches per region respectively.

3. Sample Level Scores: The model was applied to individual patches from a sample (e.g. a slide or a TMA punch) and the mean score across all patches in the slide was reported.

## Comparator models

We compared our mixed model, which combines a CD31 and Angioscore arm, to the corresponding single modality models across our three RNA cohorts (Supplementary Table 3):

1. CD31 Only Model: The segmentation model (UNet model based on ResNet18 encoder) was trained using only the CD31 mask + H&E image pairs (from the UTSW CD31 Re-stain dataset) and the model performance was calculated for the TCGA held out data as well as UTSeq and IMmotion150 data. For each slide, the fraction of pixels called as CD31 positive was correlated with RNA Angioscore.

2. Angioscore Only Model (end-to-end): It has a ResNet-18 network (pretrained on Imagenet) with the last fully connected network predicting a single value instead of 1000 classes. It was trained alone using the TCGA training dataset with the BatchMSE loss.

3. Angioscore Only Model (last layer trained): We use the same network and training as in case b., with the exception of freezing all the layers except the fully connected layer.

4. Angioscore Only Model (last two layers trained): We use the same network and training as in case b., with the exception of freezing all the layers except the last Convolution and fully connected layer.

5. CAMIL regression model: This is the Contrastively Clustered attention-based multiple instance learning model as described by

Nahhas et al.[23] Default parameters were used except for two changes which both led to improved performance: i) A ReNet50 encoder pretrained on Imagenet was used to extract features on 20X images (as opposed to RetCCL on downsampled images) and ii) Patches sampled from the tumor region as described above rather than from a grid across all tissue areas.

## Survival analysis

1. Cox-Proportional Hazards: Patients were stratified into groups based on a given threshold level for the H&E DL Angioscore (or RNA/CD31 for IMmotion150). We chose different Time variables based on the dataset, TCGA: Overall Survival, UTSW TKI Response: Time to Next Treatment, IMMmotion150: Progression Free Survival. A Univariate Cox proportional hazard model was then used to determine the characteristics associated with overall survival. Kaplan–Meier curves were generated using the lifelines python package.

2. Optimal H&E DL Angioscore threshold: Overall survival data for TCGA was used and the H&E DL Angioscore threshold was varied and overall survival outcomes for the two groups (above and below the threshold) were calculated. We sought to select the threshold with the lowest $p$ value and high hazard ratio, but found two peaks in the TCGA data, which we then used to stratify patients into Low/Medium/High H&E DL Angioscores. The same thresholds were applied across all other cohorts.

3. Comparison to standard clinical measures: For c-index calculations all categories were preserved (e.g., 4 grade levels) and in situations when the categories were categorical e.g. (MSKCC has favorable/intermediate/poor) these were transformed to numerical values (e.g. 1/2/3) before c-index calculation.

## Statistics and reproducibility

Comparisons of different measures of angiogenesis such as the H&E DL Angioscore, RNA Angioscore and CD31 were performed using the spearman rank correlation, and the corresponding $p$-values were calculated using the two-sided test spearman rank correlation test. Change in H&E DL Angio score with grade, stage and mutation status on the TMA data we quantified using Mann–Whitney U test, a non-parametric two-sided test. All survival data (overall survival for TMA data, time to next treatment for UTSW TKI response dataset and progression free survival data from IMM150) were analyzed using the two-sided non-parametric log rank test.

From TCGA data we excluded some slides from our analysis: 14 exhibited frozen sample-like artifacts, 5 had non-ccRCC-like pathology, 2 had only benign renal parenchyma, 2 slides had imaging/staining artifacts, 6 slides were duplicates from the same patient, and 28 lacked matching RNA data in the pan-cancer dataset. The TMA dataset is a combination of two existing cohorts, which had 6 patients in common. For these common patients we used the information from the cohort which provided longer survival follow up. In addition, 172 punches were excluded from the TMA dataset due to one of the following reasons: normal tissue, having predominantly artifacts, not enough tumor present, not enough or no tissue present, and having non ccRCC tissue. No data were excluded from analyses from the UTSeq dataset. In the UTSW TKI response dataset, data for patients whose treatment was stopped due to toxicity was excluded. Thirteen slides from IMM150 data were excluded due to duplicates ($n = 5$) or quality issues (significant artifacts resulting in extremely low evaluable tumor fraction). All training and validation for the TCGA dataset and the UTSW CD31 Re-stain datasets were randomly split, and no statistical test was used to determine the split sizes.

## Reporting summary

Further information on research design is available in the Nature Portfolio Reporting Summary linked to this article.

## Data availability

H&E images for TCGA KIRC can be downloaded from the TCGA GDC portal, while the corresponding gene expression data is available from cBioPortal. H&E images for the UTSW TMA cohort can be downloaded from https://doi.org/10.25452/figshare.plus.19324118. Other UTSW data used in this study can be obtained by direct email to the corresponding authors and are subject to institutional permission and ethics compliance. Data can only be shared for non-commercial academic purposes and will require a data user agreement. IMmotion150 data, including H&E images, Response, RNA Angioscores and CD31 levels is proprietary to Roche. The anonymized genomic data from 163 patients who granted informed consent to share such data, are made available by Roche at the European Genome-Phenome Archive (EGA) under accession number EGAS00001002928. Summarized figure source data are provided, except PFS data in 4B which cannot be shared in order to ensure the highest patient-level privacy. Source data are provided with this paper.

## Code availability

Code is available as a repository on GitHub https://github.com/Rajaram-Lab/dl-angio-from-hne where it can continue to be updated, as well as a frozen version at Zenodo https://doi.org/10.5281/zenodo.14783456. The H&E DL Angio model is available on Zenodo https://doi.org/10.5281/zenodo.14008281.

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

## Acknowledgements

We thank all the patients who provided tissues that enabled this research project. This grant was funded by DOD (KC200285) and in part CPRIT (RP220294) and a Pilot grant from the Lyda Hill Department of Bioinformatics. J.B., A.C., D.R. and P.K. are supported by NIH (Specialized Program in Research Excellence in Kidney Cancer P50 CA196516), J.B., A.C., and P.K. by the Cancer Research & Prevention Institute of Texas (CPRIT; RP180192). J.B. and A.C. are also supported by CPRIT (RP180191). P.K. is supported by NIH (R01CA244579, R01CA154475, and R01DK115986), DOD (KC200294), and CPRIT (RP200233). S.R. is supported by CPRIT (RP220294), DOD (KC200285) and startup funds provided through the Lyda Hill Department of Bioinformatics.

## Author contributions

Conceptualization, S.R., P.K., and J.B.; computational analysis - J.J., V.J., N.B., S.R; histology and phenotype characterization, P.K., V.P.; sample acquisition and nucleic acid extraction, D.C., J.M., D.R.; sequencing data, M.H., Z.M.; genomic/transcriptomic analysis, H.Z., Z.M.; immunohistochemistry: J.M., D.C., and P.K.; data curation, E.EKIII, J.J, V.J., A.C., N.B., P.K., and S.R.; writing – original draft: J.J, P.K., S.R.; writing – review &

editing, E.EKIII, J.B., S.R., and P.K.; resources, D.R., Z.M., S.R., P.K., and J.B.; supervision, S.R, P.K., Z.M., and J.B.; funding acquisition, S.R. and P.K.

## Competing interests

The authors declare no competing interests.
