## [Transparent Peer Review file · Nature Communications]

Histopathology Based AI Model Predicts Anti-Angiogenic Therapy Response in Renal Cancer Clinical Trial

Corresponding Author: Dr Satwik Rajaram

Version 0:

Reviewer comments:

Reviewer #1

(Remarks to the Author)

The authors present a method to estimate the angioscore directly from H&E WSI alongside CD31 IHC. They demonstrate their method is able to accurately predict angiogenesis in a range of cohort and clinical trial data.

Their findings are robust and I think demonstrate a clear value-add of machine learning applied to WSI in clinical diagnosis.

I have a few minor comments I would like the authors to address:

1. The authors state there are no works that provide both WSI gene expression prediction / trait prediction + interpretability. This is not true, as our own work Cisternino et al, 2023 demonstrates we can predict the RNA expression of thousands of genes accurately and spatially resolve them on H&E WSI, making the model directly interpretable.
2. The authors first segment CD31+ regions using image registration and then training a U-net model to perform binary segmentation. How do the results change if this step is skipped? I would assume the angioscore predictive model would just learn to focus on regions with angiogenesis / high CD31 positivity anyway?
3. Why did the authors not consider Multiple instance learning regression as a natural solution to get both tile-level angioscores and slide level angioscore averages? This seems like the most natural approach to solve the problem.
4. The authors don't report a cross-validation standard error of predictive performance, making it harder to assess the model stability given different test sets.
5. The description of both the model architecture and loss function could be written in a more clearer way. It would be nice if the model architecture was supplied as a supplementary table / figure
6. The reviewer would appreciate the code being shared on Github already for evaluation.

(Remarks on code availability)

The code is not available as the github repository is not accessible.

Reviewer #2

(Remarks to the Author)

Predictive biomarkers of treatment response are an unmet need for metastatic clear cell renal cell carcinoma (ccRCC), a tumor type for which patients may be treated with different targeted therapies including angiogenesis inhibitors, immune checkpoint inhibitors, mTOR inhibitors and/or HIF2 inhibitor. The Angioscore, an RNA-based quantification of angiogenesis, is a candidate biomarker to predict response to anti-angiogenics (AA) response. However implementation of gene expression profile based biomarkers in clinical practice is very challenging.

The present manuscript reports on a deep learning model able to predict, from HE histological digital slides, response to

anti-angiogenics by combining prediction of tumor vasculature and gene expression.

The manuscript is well written and interesting, but there are however some issues

Main comments

-The approach used by the authors is interesting, however the main limitation of the work is the low c-index performance (0.66) for prediction of the subset of patients more likely to benefit from anti-angiogenics. It is very likely that such a c index value would be sufficient for clinical adoption.

-The manuscript involves many different cohorts, some of which were not treated by antiangiogenics or who did not have RNA sequencing. It is sometimes difficult to follow the workflow, a flow chart figure would be helpful

-Authors mention the potential issue of heterogeneity : did they compare several samples from the same tumor ? or metastasis vs primary ?

-what is the meaning of « noisy » in the introduction ? could authors develop the advantages and limitations of the CD31 immunostaining

-The role of the nucleolar grade is discussed, and authors report a relationship between the nucleolar grade and the HE DL Angioscore. So maybe the nucleolar grade could be sufficient to identify the responders to antiangiogenics?

-do authors have data regarding progression free survival ?

-what happens if predictions of CD31 and RNA are not consistent ?

-the performance of the model should be confronted to that of other simple classical clinical and biological features

(Remarks on code availability)

Version 1:

Reviewer comments:

Reviewer #1

(Remarks to the Author)

The authors have addressed all my concerns - In particular the MIL comparison, SE estimates of performance and sharing of code, vastly improve the manuscript and I recommend its publication.

(Remarks on code availability)

Reviewer #2

(Remarks to the Author)

i am happy with the revisions

(Remarks on code availability)

RESPONSE TO REVIEWERS' COMMENTS

Reviewer #1 (Remarks to the Author): computational expertise in deep learning and histopathology

The authors present a method to estimate the angioscore directly from H&E WSI alongside CD31 IHC. They demonstrate their method is able to accurately predict angiogenesis in a range of cohort and clinical trial data.

Their findings are robust and I think demonstrate a clear value-add of machine learning applied to WSI in clinical diagnosis.

I have a few minor comments I would like the authors to address:

We thank the reviewer for the positive comments and constructive feedback

1. The authors state there are no works that provide both WSI gene expression prediction / trait prediction + interpretability. This is not true, as our own work Cisternino et al, 2023 demonstrates we can predict the RNA expression of thousands of genes accurately and spatially resolve them on H&E WSI, making the model directly interpretable.

This is an excellent point. It was remiss of us to not discuss the ability of multiple instance learning (MIL) regression approaches – by far the most common way of approaching this type of problem – in providing patch level scores towards interpreting a sample level result.

While one might conceivably gain insights about the model by comparing multiple patches with low and high scores, we believe that being able to visualize the specific pixels which drive the predictions provides a significantly higher level of direct interpretability. For example, this could directly tell us whether vascular boundaries are too thick, stroma tissue are incorrectly being called vascular and so on. For this reason, pixel-level interpretability was a design principle of our approach.

We thank the reviewer for pointing out this omission, and have attempted to convey our message more clearly by:

- 1. Adding an explicit contrast between interpretability of MIL and our approach in the introduction and discussion.
- 2. Modifying Fig 1C, to highlight the difference between patch level and pixel level predictions so our point is more visually apparent.
- 3. We added more references related to MIL regression models that have been published since we first submitted this manuscript.

2. The authors first segment CD31+ regions using image registration and then training a U-net model to perform binary segmentation. How do the results change if this step is skipped? I would assume the angioscore predictive model would just learn to focus on regions with angiogenesis / high CD31 positivity anyway?

3. Why did the authors not consider Multiple instance learning regression as a natural solution to get both tile-level angioscores and slide level angioscore averages? This seems like the most natural approach to solve the problem.

As pixel level interpretability was a design principle of our approach (even potentially at the cost of raw predictive power), we initially did not do extensive comparisons to such other approaches. However, as the reviewer rightly points out, given the non-traditional nature of our approach, readers may be interested in the impact of various machine learning choices. Compared to the usual MIL approach our model used RNA and CD31 data, performs end-to-end training and subsamples a fraction of patches from a slide in every batch. Accordingly, we compared the following models in their ability to predict the RNA angio score in our three cohorts:

1. Our full mixed model with both CD31 and Angio arms. This model uses end to end training, and the Batch MSE loss (based on sampling a subset of patches per slide in a batch) for the Angio arm and the usual segmentation loss for the CD31 arm.

2. An Angio only model based on a traditional Resnet18 network with the Batch MSE, and with one of a) end-to-end training b) last two layers trained and c) last layer trained.

3. The CD31 only model trained end-to-end with corresponding segmentation loss.

4. MIL (using the recently published CAMIL framework by Nahhas et al.) where pre-extracted features are used from a much larger number of patches per slide. We note that we initially got poor performance using the defaults from this approach, possibly due to sampling from all areas and not just tumor regions as we do and down sampling patches at MPP (micros per pixel) values of 1.14, prior to obtaining features with RetCCL. We achieved significantly improved performance using our tumor-only patches and a more comparable Resnet based backbone, which is what we report.

Across all three dataset (Supp Table 3 and updates to results/discussion) we found that the mixed model consistently outperformed the single arm models and the MIL model. The MIL model performed very similarly to the Angio only model with only the last layer trained, even though the latter sees a far smaller number of patches per slide in a batch. We expect this result is somewhat specific to vasculature which is well represented by a small selection of patches: it shows regional variation but is reasonably homogenous (and in contrast with say immune cell prediction which in our experience needs dense tumor sampling). Interestingly, the Angio only models with more weights trained perform better on (held out) TCGA but generalize worse to other datasets suggesting overfitting which the inclusion of the CD31 arm in the mixed model is able to rectify. It is of course conceivable that a better encoder, or different modeling choices, could lead to improved MIL performance, however this alone would not enable pixel-level-interpretability which we considered a requirement.

4. The authors don't report a cross-validation standard error of predictive performance, making it harder to assess the model's stability given different test sets.

We have now performed a "cross-validation" type training using three folds of TCGA data to train three separate models, with the first best performing (on TCGA) fold being our original model. These three models were applied across various downstream tasks, and we show the prediction on RNA (e.g., correlation with UT Seq = 0.77/0.76/0.74, correlation with IMM150 = 0.73,0.73,0.72) and response (c-index on IMM150 0.66/0.63/0.64) are largely consistent. We added additional table (supplementary Table 2) with Cross validation results.

5. The description of both the model architecture and loss function could be written in a more clearer way. It would be nice if the model architecture was supplied as a supplementary table / figure

We have added a new supplementary figure 1 b explaining the BatchMSE (Batch Mean Square Error) loss to supplement the existing Supp. Figure 1 which explained the model. We have also revised the description in the methods to improve clarity.

6. The reviewer would appreciate the code being shared on Github already for evaluation.

Reviewer #1 (Remarks on code availability):

The code is not available as the github repository is not accessible.

We have made the code publicly available on GitHub in addition to the previously shared Supplementary file.

Reviewer #2 (Remarks to the Author): expertise in HCC pathology

Predictive biomarkers of treatment response are an unmet need for metastatic clear cell renal cell carcinoma (ccRCC), a tumor type for which patients may be treated with different targeted therapies including angiogenesis inhibitors, immune checkpoint inhibitors, mTOR inhibitors and/or HIF2 inhibitor. The Angioscore, an RNA-based quantification of angiogenesis, is a candidate biomarker to predict response to anti-angiogenics (AA) response. However, implementation of gene expression profile based biomarkers in clinical practice is very challenging.

The present manuscript reports on a deep learning model able to predict, from HE histological digital slides, response to anti-angiogenics by combining prediction of tumor vasculature and gene expression.

The manuscript is well written and interesting, but there are however some issues

We thank the reviewer for their positive feedback.

Main comments

-The approach used by the authors is interesting, however the main limitation of the work is the low c-index performance (0.66) for prediction of the subset of patients more likely to benefit from anti-angiogenics. It is very likely that such a c-index value would be sufficient for clinical adoption.

We broadly agree with the commonly held notion that a c-index of greater than 0.7 is desirable for clinical use (which we note we are already close to). However, as detailed below, the relevant context is improvement over current practice.

1. Currently there are no guidelines for making the decision to treat RCC patients with anti-angiogenic therapy, and treatments are often determined based on Favorable/Intermediate/Unfavorable assessments based on the IMDC or MSKCC risk scores. In the IMM150 data the c-index for the MSKCC (IMDC not available) is 0.53 for the sunitinib arm which is much lower than the 0.66/0.67 for the H&E/RNA based angioscores.

2. Though for a different task, current widely used clinical risk models such as tumor, node, metastasis (TNM) staging model and size, stage, grade, and necrosis (SSIGN) score for postoperative prognostication also do not meet the 0.7 threshold. Correa et. al. [PMCID: PMC7085167] independently evaluate 8 clinically relevant recurrence models on an independent cohort of 1647 RCC patients from the ASSURE clinical trial. They report that using TNM staging model as a reference (C-index = 0.6), the most accurate prognostic model was the SSIGN score, and with a c-index of 0.688, it barely exceeded the c-index we observe (albeit on a different task) from our model.

3. We imagine that for practical application, much as with the IMDC risk score, the ultimate decision-making criterion will be a nomogram considering multiple factors. This notion is supported by data from IMmotion151 suggesting that the typically used prognostic parameters as well as other readouts like PD-L1 offer independent information to the RNA based signatures [PMCID: PMC8436590]. While we do not have the training data to reliably train/test a nomogram, we performed an extremely naïve ranked averaging of the H&E DL Angio Score with various clinical

readouts and were able to bump the c-index up to 0.68 for the IMmotion150 data. Any trained model would be expected to perform significantly better.

Parameter Used	C-Index
PDL1_TC_Category Alone	0.57
Motzer Group Alone	0.58
H&E Model Alone	0.66
Motzer Group + H&E Model	0.66
PDL1_TC_Category+H&E Model	0.68

4. Heterogeneity: We discuss this further below, but a primary advantage of our model is the possibility of profiling multiple areas in a tumor to overcome intra-tumor heterogeneity. This is the focus of an ongoing project and we expect being able to faithfully sample whole tumors will further improve the prediction of response.

We now provide this information greater detail in our discussion to help readers better contextualize our contribution

-The manuscript involves many different cohorts, some of which were not treated by antiangiogenics or who did not have RNA sequencing. It is sometimes difficult to follow the workflow, a flow chart figure would be helpful

This is an excellent suggestion. We have now added a new supplementary table 1 to compare our various cohorts.

-Authors mention the potential issue of heterogeneity : did they compare several samples from the same tumor ? or metastasis vs primary ?

This is a great question. IMmotion150 only had a single slide per sample, so we were not able to test the role of heterogeneity in response prediction there. Accordingly, this is the focus of our ongoing efforts and we fully expect to get a significant bump to the c-index when considering multiple samples per patient. That said to highlight the prevalence of intratumor heterogeneity (ITH) in angioscore (ccRCC is the paradigm for genetic ITH) for the reader we have a) Added a reference (PMCID: PMC9762114 Golkaram et. Al. 2022), where authors show that angiogenesis significantly

varied across tumor regions within the same patient and b) further highlighted Fig 1C where we show multiple regions in a slide differing Angiogenesis

-what is the meaning of « noisy » in the introduction ? could authors develop the advantages and limitations of the CD31 immunostaining

We have updated this language in the introduction to clarify that we mean that the CD31 expression only weakly correlated with response and showed much variation between patients.

-The role of the nucleolar grade is discussed, and authors report a relationship between the nucleolar grade and the HE DL Angioscore. So maybe the nucleolar grade could be sufficient to identify the responders to antiangiogenics?

-the performance of the model should be confronted to that of other simple classical clinical and biological features

We agree we should have done a better job in clarifying how our model improves over currently available variables. On the IMmotion150 dataset we repeated our analysis based on various clinical variables available to us (Supp Table 5) and show that indeed these variables have much lower performance (c-index <0.6).

-do authors have data regarding progression free survival ?

We do use Progression Free Survival for IMmotion150. However, on the TMA survival results we do not have this information. For the UTSW patients that received drug therapy, we have data for time to next treatment which we thought was a more relevant readout for treatment efficacy.

-what happens if predictions of CD31 and RNA are not consistent ?

CD31 and RNA are used in multiple contexts, and we were unsure which the reviewer was referring to:

1. If by predictions of CD31 and RNA they are referring to the two arms of the model. We have shown that these are very highly correlated. Ultimately for interpretability we chose to rely on the arm that predicts the vascular network.
2. If they mean the ground truth data: Although our results show that CD31 and RNA are correlated, there are clear differences in the quality (possibly due to the points raised above) between the two. Specifically, it is clear both from the IMmotion 150 data and our results, that the RNA data is a more reliable predictor of treatment response.
3. Finally, if they are asking about the situations where our models predictions diverge from the RNA: We have reviewed these carefully but haven't found a clear pattern one way or the other. For example, in one instance that our model and pathologist clearly thought was low vasculature, the RNA gave a high angio score, we found that there was vasculature just outside the tumor area, which we surmised must have been captured during macrodissection.